# Glucocorticoid Resistant Pediatric Acute Lymphoblastic Leukemia Samples Display Altered Splicing Profile and Vulnerability to Spliceosome Modulation

**DOI:** 10.3390/cancers12030723

**Published:** 2020-03-19

**Authors:** Rocco Sciarrillo, Anna Wojtuszkiewicz, Irsan E. Kooi, Leticia G. Leon, Edwin Sonneveld, Roland P. Kuiper, Gerrit Jansen, Elisa Giovannetti, Gertjan J.L. Kaspers, Jacqueline Cloos

**Affiliations:** 1Amsterdam UMC, Vrije Universiteit Amsterdam, Departments of Pediatric Oncology, Hematology and Medical Oncology, Cancer Center Amsterdam, 1081 HV Amsterdam, The Netherlands; 2Amsterdam UMC, Vrije Universiteit Amsterdam, Departments of Pediatric Oncology and Hematology, Cancer Center Amsterdam, 1081 HV Amsterdam, The Netherlands; 3Amsterdam UMC, Vrije Universiteit Amsterdam, Cancer Center Amsterdam, Department of Clinical Genetics, 1081 HV Amsterdam, The Netherlands; 4Erasmus MC, University Medical Center Rotterdam, Department of Immunology, 3000 CA Rotterdam, The Netherlands; 5Princess Máxima Center for Pediatric Oncology, 3584 CX Utrecht, The Netherlands; 6Amsterdam UMC, Vrije Universiteit Amsterdam, Amsterdam Immunology and Rheumatology Center, Cancer Center Amsterdam, 1081 HV Amsterdam, The Netherlands; 7Amsterdam UMC, Vrije Universiteit Amsterdam, Department of Medical Oncology, Cancer Center Amsterdam, 1081 HV Amsterdam, The Netherlands; 8Cancer Pharmacology Lab, AIRC Start-Up Unit, Fondazione Pisana per la Scienza, 56017 San Giuliano Terme (Pisa), Italy; 9Emma’s Children’s Hospital, Amsterdam UMC, Vrije Universiteit Amsterdam, Pediatric Oncology, 1081 HV Amsterdam, The Netherlands; 10Amsterdam UMC, Vrije Universiteit Amsterdam, Department of Hematology, Cancer Center Amsterdam, 1081 HV Amsterdam, The Netherlands

**Keywords:** alternative splicing, glucocorticoid resistance, pediatric acute lymphoblastic leukemia, RNA sequencing, SF3B modulators

## Abstract

Glucocorticoid (GC) resistance is a crucial determinant of inferior response to chemotherapy in pediatric acute lymphoblastic leukemia (ALL); however, molecular mechanisms underlying this phenomenon are poorly understood. Deregulated splicing is a common feature of many cancers, which impacts drug response and constitutes an attractive therapeutic target. Therefore, the aim of the current study was to characterize global splicing profiles associated with GC resistance and determine whether splicing modulation could serve as a novel therapeutic option for GC-resistant patients. To this end, 38 primary ALL samples were profiled using RNA-seq-based differential splicing analysis. The impact of splicing modulators was investigated in GC-resistant leukemia cell lines and primary leukemic specimens. Our findings revealed, for the first time, markedly distinct splicing landscapes in ALL samples of B-cell precursor (BCP)-ALL and T-ALL lineages. Differential splicing events associated with GC resistance were involved in RNA processing, a direct response to GCs, survival signaling, apoptosis, cell cycle regulation and energy metabolism. Furthermore, our analyses showed that GC-resistant ALL cell lines and primary samples are sensitive to splicing modulation, alone and in combination with GC. Together, these findings suggest that aberrant splicing is associated with GC resistance and splicing modulators deserve further interest as a novel treatment option for GC-resistant patients.

## 1. Introduction

Improvements achieved over the years in the treatment of childhood acute lymphoblastic leukemia (ALL) have resulted in five-year survival rates approaching or even exceeding 90% in some high-income countries [1,2]. However, up to 20% of patients experience a relapse which is largely caused by therapy resistance of leukemic cells and comes with a dismal prognosis. Therefore, further research on the identification and molecular characterization of drug-resistant subsets of ALL is warranted [3,4,5,6].

ALL patients are treated with combination chemotherapy, including glucocorticoids (GCs), in particular dexamethasone (Dex) and predniso(lo)ne (Pred), which are administered at induction the phase together with vincristine, L-asparaginase and anthracyclines [2,7,8]. Importantly, in vitro resistance of primary childhood ALL cells to Pred was shown to correlate with both diminished short and long-term clinical response to chemotherapy [9,10]. Moreover, blasts isolated at relapse displayed higher GC resistance as compared to leukemic cells at diagnosis, which further underscores the importance of GCs in ALL treatment outcomes [11].

Mechanistically, GC resistance frequently arises from the impaired activity of the glucocorticoid receptor (GR) caused by diminished gene expression, mutations and reduced ligand and/or DNA binding capacity [8]. GC response is mediated by various signaling transduction pathways downstream the GR, including MAPK or NFκB signaling [12,13,14], and is suppressed by increased levels of anti-apoptotic proteins such as BCL2 and MCL1 [15,16,17,18]. Moreover, alterations in transcription factors interacting with GR affect its ability to regulate the expression of target genes [8,18]. For instance, loss of IKZF1 function via gene deletions and splicing alterations was shown to directly alter GC transcriptional response and induce GC resistance in B-cell-precursor (BCP) ALL [19,20]. In a genome-wide effort to identify the genetic basis of drug sensitivity in ALL, Holleman et al. [15] revealed global gene expression patterns associated with resistance to the core chemotherapeutics in ALL treatment, including Pred. Differentially expressed genes between GC-resistant and sensitive ALL samples involved multiple cellular processes including carbohydrate metabolism, apoptosis and nucleic acid metabolism. Although many studies demonstrated the association of differential gene expression with drug resistance in ALL [15,21,22], alterations in global mRNA splicing profiles have never been investigated in this context.

Deregulated splicing occurs in many cancers and plays an important role in hematological malignancies. Mutations in splicing factors influence pathogenesis and drug resistance in myelodysplastic syndromes (MDS), acute myeloid leukemia (AML) and chronic lymphocytic leukemia (CLL) [23,24,25,26,27,28]. In these entities, splicing factor mutations perturb global splicing profiles, thereby creating vulnerability to drugs targeting the spliceosome and, in particular, the SF3B complex [29,30]. Beyond splicing factor mutations, Yang and collaborators recently demonstrated that high levels of global aberrant splicing have a negative prognostic impact in MDS patients [31].

Whereas spliceosome mutations were infrequent in ALL [32], BCP-ALL cells did display global aberrant splicing when compared with non-malignant controls [33,34]. Interestingly, we previously showed that aberrant splicing of folylpolyglutamate synthetase (FPGS), a key determinant of methotrexate (MTX) efficacy, is associated with MTX resistance in childhood ALL [35,36]. Notably, high levels of one particular aberration, i.e., FPGS intron 8 partial retention, were also associated with increased resistance to GCs. This suggests that leukemic cells of GC-resistant patients carry a more profound splicing dysregulation which could be exploited for therapeutic purposes. 

Building on these data, the aim of the current study is to characterize the global alternative splicing profiles associated with ex vivo GC resistance in childhood ALL. Specifically, we investigated differential splicing profiles in 38 primary childhood ALL samples by using RNA sequencing [37]. Finally, we tested whether resistant cell lines and primary specimens can be sensitized to GC treatment by using SF3B modulators.

## 2. Results

### 2.1. Differential Splicing Landscape Associated with GC Resistance in Pediatric ALL

In order to evaluate whether GC resistance is associated with specific splicing patterns in childhood ALL, we used RNA sequencing to profile transcriptomes of specimens obtained from 36 newly diagnosed and two relapsed pediatric ALL patients (Figure 1A). This study cohort was well characterized with respect to clinical features and ex vivo drug resistance, including Dex and Pred (Appendix A). The patient specimens were classified either as GC-sensitive (*N* = 15) or GC-resistant (*N* = 23) based on ex vivo Dex and Pred LC_50_ values according to the previously established cut-offs of 0.01 µg/mL and 0.1 µg/mL, respectively [9] (Figure 1B). Furthermore, we determined the immunophenotype and genetic profile of the samples, including mutations in GR and recurrent genetic alterations associated with ALL [21,38] (Figure 1C), which allowed us to account for possible confounders in the analysis.

The global differential splicing profiles of GC-sensitive and GC-resistant samples were determined using the rMATS algorithm (Figure 2A). This software identifies sequencing reads which support a certain splice event (e.g., the inclusion or skipping of a certain exon in a gene of interest) and calculates the inclusion levels (or Percentage Spliced-In, Ψ). Ψ is computed as the proportion of reads supporting the inclusion of the exon in question divided by the sum of reads supporting the inclusion and skipping of this exon. Subsequently, it compares the average Ψ values of GC-sensitive specimens with that of GC-resistant samples by computing the Inclusion Level Difference (∆Ψ) and the corresponding *p*-value and false discovery rate (FDR) for each splice event [37]. We found, in total, 994 significant differential splicing events (FDR < 0.05, Figure 2B and Appendix A) affecting 762 genes. Hierarchical clustering (Figure 2C) and principal component analysis (Figure 2D) based on Ψ (inclusion level) values of the detected splicing events showed that the majority of GC-resistant and GC-sensitive samples clustered together; however, five specimens from each group clustered in their opposing class.

### 2.2. BCP- and T-Cell ALL Show Distinct Splicing Patterns

Previous reports indicate that BCP-ALL harbored distinct gene expression signature compared to T-ALL [15], which was confirmed in the current dataset (Figure A1A and Appendix A). Since alternative splicing is known to demonstrate tissue-specificity [39,40], cell-type-specific differences could potentially affect our analysis. To address this issue, we examined whether the differences between BCP-ALL and T-ALL were also reflected in splicing profiles. By comparing the transcriptomes of six T-cell and 26 BCP (pre-B and common) ALL samples with rMATS, we found 2097 significant differential splicing events (Figure A1B,D) affecting 1416 genes.

Gene ontology (GO) analysis (Figure A1E) revealed that genes differentially spliced between BCP and T-ALL are largely involved in the regulation of transcription and histone modifications, mRNA processing as well as several signaling pathways. Not surprisingly, many differentially spliced genes were involved in B- and T-cell differentiation processes (i.e., LCK, ICAM2, RUNX1), including T-cell receptor signaling and MAPK cascade (i.e., FLT3, STAT5B and MAP kinases) (Appendix A). Given that both subtypes of ALL show fundamental differences in their splicing profiles as well as in previously reported mechanisms of resistance to GCs [19,41], we proceeded to analyze differential splicing in relation to GC resistance within each subtype separately.

### 2.3. Differential Splicing in Relation to GC Resistance in BCP-ALL and T-ALL

Differential splicing analysis performed on 10 GC-sensitive and 16 GC-resistant BCP-ALL samples uncovered 1035 significant events affecting 777 genes and resulting in a better separation of the two groups compared to the total cohort (Figure 3A,B). The majority of splicing events in the BCP-ALL had ∆Ψ ranging between –0.05 and 0.05, indicating overall small differences between GC-resistant and GC-sensitive samples (Appendix A). The comparison between two GC-sensitive and four GC-resistant T-ALL samples revealed a total of 932 significant differential splicing events occurring in 722 genes (Figure 3A,B and Appendix A).

Our unbiased GO analysis revealed that some processes are commonly affected in both BCP-ALL and T-ALL (Figure 3C,E). This includes for instance factors involved in RNA metabolism, in particular regulation of transcription (histone modifications) and mRNA splicing (i.e., U2AF1, multiple HNRNPs and DDX helicases). Interestingly, perturbation of some members of this commonly affected pathway was subtype-specific, as SRSF3 and SRPK2 were found alternatively spliced in the T-ALL dataset only, while SRSF5 and SRSF7 were specific for the BCP-ALL group. Furthermore, many processes directly linked to GC-induced responses were affected in a subtype-specific manner.

Interestingly, alternative splicing in BCP-ALL also affected multiple genes encoding proteasomal subunits which regulate important B-cell functions, including antigen presentation, NFkB and Wnt signaling pathways. Furthermore, this dataset was enriched in genes regulating apoptosis and cell cycle (Figure 3D) which can influence GC resistance. In particular, the pro-apoptotic factor BAX was affected by increased exon skipping (ES) in GC-resistant samples. TP53 isoform γ (exon 9γ inclusion) appeared more expressed in sensitive samples, while TP53 isoform β (exon 9β inclusion) showed higher expression in resistant samples. 

Superoxide dismutases (SOD1, cytoplasmic and SOD2, mitochondrial) encode antioxidant enzymes that degrade harmful superoxide radicals in the cell. Both SOD1 and SOD2 showed an increase in exon inclusion in GC-resistant samples. Furthermore, differential splicing affected two genes directly linked to GR signaling: HSP90AA1, GR-associated molecular chaperone, and SGK1, serum/glucocorticoid regulated kinase 1. The latter plays an important role in cellular responses to stress and exerts anti-apoptotic functions [18]. In addition, we identified a group of genes involved in focal adhesion and cytoskeleton organization which holds importance for cell-stroma interaction and the acquisition of stem-like properties (i.e., VIM, CD44, ITGA4/5). 

Multiple genes involved in energy metabolism were differentially spliced uniquely in the context of GC resistance in T-ALL (Figure 3E,F). This included regulatory glycolytic enzymes (PFKL and PFKM), core members of the ubiquinol-cytochrome c reductase (UQCRC1 and UQCRC2) as well as five crucial subunits of the NADH dehydrogenase (NDUF, Figure 3F). These components of energy metabolism were affected by RI events elevated in GC-resistant samples, as well as alternative 3’ or 5’ splice site selection (ASS3 and ASS5).

In addition, a more targeted search within the T-ALL subtype revealed differentially spliced genes involved in several signaling pathways crucial to cell survival (Figure 3F). This included BAX (affected by increased RI in GC-resistant samples), members of the NFκB (i.e., CHUK showing increased inclusion of exon 7 in GC-resistant samples and IKBKB with a minor A5SS event) and MAPK signaling (i.e., MAP2K2 showing a very minor RI and MAP3K8 displaying ES in GC-resistant samples).

Finally, we evaluated whether splicing perturbations of genes involved in pathways relevant for GC resistance directly affected sequences coding for structural and/or functional domains (Figure A2A,D, Appendix A). This was indeed observed for apoptosis and cell cycle regulators in BCP-ALL and suggests that these events can have a negative impact on protein function. In contrast, many factors involved in energy metabolism and signaling in T-ALL were not directly affected by splicing aberrations within specific protein domains.

### 2.4. Validation of Selected Differential Splicing Events

To validate the results of differential splicing analysis, we carried out semi-quantitative RT-PCR to confirm 13 splice events in BCP-ALL and 19 splice events in T-ALL, in which was found a significant differential between GC-resistant and GC-sensitive samples by rMATS (Figure 4). The PCR-based ratios were calculated analogously to the Ψ values. The correlation between rMATS and RT-PCR results varied for different genes, partly because ∆Ψ values were often relatively small between the two groups, despite being significant. However, there was a moderate to a strong statistically significant correlation between the rMATS-generated data and RT-PCR. For all 13 genes in BCP-ALL: *R*^2^ = 0.602, *p* < 0.01. For all 19 genes in T-ALL: *R*^2^ = 0.4717, *p* < 0.01 (Figure A3).

### 2.5. Splicing Modulation Potently Inhibits Growth of Dex-Resistant ALL Cells and Co-Operates with Dex to Eradicate T-ALL Cells

To evaluate the therapeutic potential of splicing modulation in drug-resistant ALL, we first determined the response to pladienolide B (Plad-B) in a T-ALL cell line CCRF-CEM-WT and three Dex-resistant CEM sublines: CEM/R30dm (unknown mechanism of Dex resistance), CEM-R5 and CEM-R5C3 (both characterized by defective GR function). Remarkably, all cell lines responded to comparably low nanomolar concentrations of Plad-B (Figure 5A). This growth inhibition was associated with dose and time-dependent cell cycle arrest (Figure 5B,C). RNA sequencing performed on CEM-WT and CEM/R30dm cells treated with 4nM Plad-B for 6 h revealed wide-spread changes in splicing (Figure 5D).

To further validate these results, we tested the sensitivity to the splicing modulators Meayamycin B (MAMB) and Plad-B in primary cells of 17 childhood ALL samples and 11 non-malignant bone marrow controls using MTT assay. Notably, primary ALL samples showed a remarkable sensitivity to both agents (Figure 5E) and the LC_50_ values in ALL cells tended to be lower compared to non-malignant samples (*P* = 0.07, median LC_50_ values 0.42 ± 0.05 nM and 0.57 ± 0.1 nM, respectively, for MAMB and *P* = 0.04, mean LC_50_ values 9.9 ± 2.1 nM and 18.7 ± 3.2 nM, respectively, for Plad-B).

To gain further insight into the potential therapeutic window of these compounds, we exposed 7 T-ALL samples to Plad-B for 72 h, followed by flow cytometry-based immunophenotyping (Figure 5F). We observed that, although in two samples, mature CD5+ T-cells were less affected as compared to blast cells (also see Figure 5G), overall, both the blasts and mature T-cells were affected to a similar extent. In contrast, CD19+ B-cells were significantly less affected than T-lineage cells (*P* = 0.04, Figure 5F).

Next, we tested whether Plad-B combined with Dex is more selective towards leukemic cells as compared to Plad-B alone. For this analysis, we selected T-ALL samples characterized by mild to high Dex resistance. To this end, we treated five samples of T-ALL patients for 72 h with the combination of Plad-B and Dex, as well as each of the drugs alone. In four cases, we observed that leukemic blasts were more efficiently eradicated by the combination as compared to both single drugs (Figure 5G). Interestingly, leukemic blasts were affected by the combination to a larger extent when compared to mature lymphocytes (CD5+ T-cells and CD19+ B-cells); however, there was some variation between patients.

Finally, we assessed the combination of 1 nM Plad-B with a range of Dex concentrations in our Dex-resistant CEM cell line models. While in CEM-R5 and CEM-R5C3 the two agents showed slightly antagonistic interaction (not shown), in CEM/R30dm the combination was highly synergistic (mean CI = 0.276 ± 0.055, Figure A4). This suggests that patients with non-GR-related Dex resistance could potentially benefit from this drug combination.

## 3. Discussion

To our knowledge, this is the first study to show the association between ex vivo GC resistance and altered splicing profiles in pediatric BCP- and T-ALL. The detection of differential splicing in primary samples could potentially be affected by large intra-patient heterogeneity of clinical and genetic features. The patient groups compared in our study were homogeneous, except for CDKN2A/B, BTG1 and EBF1 alterations which appeared more frequently among GC-resistant samples. However, we did not observe distinctive splicing profiles associated with any of these genetic aberrations.

We report that differences between BCP- and T-ALL cells at the gene expression level are also extended to alternative splicing. Many of the differentially spliced genes between BCP and T-ALL were involved in processes related to normal biological/immune function of both subtypes (i.e., B-cell and T-cell differentiation including T-cell receptor and MAPK signaling) and, therefore, likely reflect lineage-specific functional differences [42,43]. It was previously reported that many genes are alternatively spliced in the context of normal immunological functions of healthy T-cells, including specific pathways being regulated by splicing upon activation [40,44]. This data illustrates that it is crucial to investigate GC resistance in a lineage-specific context. 

We found several pathways perturbed by differential splicing in BCP and T-ALL in relation to GC resistance. Selected splicing events, even though characterized by low ∆Ψ values, were validated by RT-PCR and showed moderate to strong correlation to rMATS-derived Ψ values in the majority of cases. However, due to the semiquantitative nature of the RT-PCR method, our results should be additionally confirmed by qRT–PCR in future studies. In our dataset, several differentially spliced genes were involved in RNA processing and splicing regulation. Some of these splice factor genes were common to both ALL subtypes (i.e., U2AF1 and HNRNPA1) while others were specific to BCP-ALL (i.e., HNRNPA2B1, HNRNPK and HNRNPM, SRSF5 and SRSF7) or T-ALL (SRSF3 and SRPK2). Such proteins are known to auto-regulate their own splicing [45] and are likely to influence global splicing profiles and thereby affect cancer pathogenesis and progression. For instance, in the context of BCP-ALL, Black and collaborators [33] reported altered splicing of several cancer driver genes to be more common compared to somatic mutations. Moreover, they report that differential splicing of HNRNPA1 - whose expression levels are altered in several cancer types - leads to transcript degradation. The differences found between ALL subtypes highlighted in our data could reflect the tissue-specificity of splicing regulation. In particular, we want to emphasize that specific classes of splicing regulators are associated with specific immunophenotypes. Hence, distinct regulatory splicing pathways/networks related to GC resistance should be investigated within each subtype, to avoid interference of different splicing-related to processes such as cell differentiation. Future studies should evaluate to which degree differential splicing of these particular splicing factors contributes to the globally altered splicing profiles in GC-resistant cells.

Leivonen et al. [46] showed that alternative splicing profiles of high-risk diffuse large B-cell lymphoma (DLBCL) patients are associated with survival, distinguish DLBCL molecular subtypes and influence genes involved in drug resistance. Extending these observations to ALL, we showed that multiple genes relevant for GC response are alternatively spliced in primary childhood ALL cells, potentially contributing to diminished susceptibility to GCs. Firstly, we found splice alterations of HSP90AA1 and SGK1, both involved in GR signaling, suggesting that aberrant splicing could indirectly affect proper GR function. Secondly, many genes involved in signaling pathways, cell cycle regulation and apoptosis (known mediators of GC signaling) were differentially spliced in the context of GC resistance. This is in agreement with previous reports showing that primary ALL cells which are in vitro resistant to GCs display defects occurring downstream of nuclear translocation of the GR [47,48]. For instance, TP53 spliced variants β and γ have previously been reported to alter the transcriptional activity of TP53α consequently affecting cell cycle and apoptosis regulation [49]. Although functional differences between TP53β and TP53γ remain unclear, they could presumably interfere with GC-related anti-proliferative and pro-apoptotic effects. GC-induced apoptosis was also reported to be antagonized by increased signaling via kinase networks, including the MAPK pathway [12,13,18]. In this study, we found GC resistance-related aberrant splicing of the pro-apoptotic protein BAX as well as genes involved in multiple signaling pathways (i.e., NFκB, MAPK, PI3K/AKT and JNK signaling). For instance, elevated exon 7 inclusion of CHUK found in GC-resistant T-ALL samples could potentially lead to increased activity of this NFκB activator and consequently increased cell survival. Similarly, differential splicing of genes coding for subunits of the proteasome, an important regulator of NFκB signaling, might contribute to GC resistance in BCP-ALL. In line with this hypothesis, primary Dex-resistant ALL specimens were previously shown to display sensitivity to proteasome inhibitors, which in low concentrations were able to sensitize these samples to Dex [50].

Interestingly, genes involved in energy metabolism were differentially spliced, particularly in T-ALL. GCs induce a metabolic shift from glycolysis towards oxidative phosphorylation and it is postulated that oxidative stress plays a central role in GC-induced apoptosis of leukemic cells [12,51]. We found that GC-resistant T-ALL cells display altered splicing (in particular intron retention) of multiple genes involved in oxidative phosphorylation, in particular components of ubiquinol-cytochrome c reductase and NADH dehydrogenase complex (which promotes DNA damage-induced apoptosis through the production of reactive oxygen species (ROS)) [51]. As intron retention often results in premature stop codons and transcript degradation, these changes are likely to result in diminished GC-induced oxidative phosphorylation and, therefore, suppress the production of ROS and consequently apoptosis induction. Similarly, we found increased inclusion of specific exons in SOD1 and SOD2 genes in resistant BCP-ALL samples. GC treatment was shown to downregulate SOD1 which renders cells more vulnerable to ROS [18]. It is conceivable that GC-resistant BCP-ALL cells are less prone to GC-induced ROS-mediated apoptosis due to an increased level of SOD1. However intriguing, these hypotheses warrant functional validation. Many of the detected splicing events directly affect sequences coding for structural and/or functional protein domains and, therefore, are likely to result in loss of protein function. The impact of alterations that do not directly affect protein domains is more challenging to predict; however, intron retentions are likely to introduce premature termination codons resulting in transcript degradation or truncated dysfunctional proteins. Furthermore, evaluation of the clinical and prognostic relevance of the splicing alterations uncovered in this study was hampered by insufficient sample numbers to be able to perform survival analysis. Low sample numbers (in particular for T-ALL) are a limitation of the current study and, therefore, our findings should be further confirmed in larger datasets. Subsequent studies should focus on determining the fate and functionality of GC-related splice variants, followed by mechanistic studies and evaluation of the clinical relevance. 

Finally, we demonstrated that splicing modulators are highly effective against (GC-resistant) ALL cells. Our data extend a previous report showing that MAMB was able to eradicate multidrug-resistant breast cancer cells [52] and provide an attractive pharmacological opportunity for chemo-refractory patients. Furthermore, our results suggest that combining splicing modulation with GCs might elicit a favorable effect in T-ALL patients with functional GR; however, the response showed variation between individuals. Possible reasons for these differences include technical issues (differences in viability of untreated cells between samples during the course of the experiment) as well as genetic differences. For instance, it has previously been reported that cells with increased MYC/MYCN expression are more vulnerable to splicing modulation [53,54,55]. In the current study, we assessed spliceosome modulation in a very small pilot sample set, which limits the conclusions that can be drawn regarding the intra-individual differences in response to Plad-B. Future investigations should extend the current findings by focusing on specific subtypes of ALL to determine whether any specific genomic or transcriptomic background renders ALL cells particularly sensitive to this novel therapeutic strategy. 

Regarding potential toxic effects of Plad-B on non-malignant cells, we observed that in primary T-ALL samples Plad-B affects mature T-cells to a similar extent as blast cells, whereas B-cells remained largely unaffected. However, considering that mature T-cells in these patients are of the same lineage as leukemic blasts, they might not completely reflect healthy cells. The notion that the combination of Plad-B with Dex had more selective effects on blast cells as compared to mature lymphocytes further encourages confirmatory studies in larger sample sets and with different classes of spliceosome modulators. Given the variable potency of such compounds, future tests in combination with GCs should be carried out to determine an optimal therapeutic window.

## 4. Materials and Methods 

### 4.1. Primary Childhood ALL Patient Samples and Leukemic Cell Lines

Thirty-six cryopreserved mononuclear bone marrow (BM) or peripheral blood (PB) cells from pediatric ALL patients at diagnosis and 2 at relapse were included in this study. Patients were treated according to the Dutch Childhood Oncology Group (DCOG) protocols ALL6–ALL9 [56] or German Co-operative ALL (Co-ALL) protocols 92–97 [3]. All patients included in the study provided written informed consent. The studies have been approved by the local medical ethical committee. Information about the patient’s karyotype, immunophenotype and ploidy were provided by DCOG.

Samples with leukemic blast content >80% were selected for MTT-based drug cytotoxicity screening, as previously described [57], and processed for RNA sequencing. Genetic alterations for IKZF1, BTG1, EBF1, PAX5, CDKN2A/B, JAK2, PAR1 and RB1 were determined through MLPA analysis as previously described [58]. The pediatric non-malignant controls were derived from 1 PB and 3 BM specimens of patients with cardiological disease and 2 specimens taken from 2 T-ALL patients in remission (<5% blasts).

The human T-ALL cell line CCRF-CEM was obtained from ATCC. Dex-resistant CEM sublines used in this study included: FPGS-deficient MTX-resistant, GC-cross-resistant subline CEM/MTX-R30dm (kindly provided by Prof. J. McGuire [59]) and CEM-R5 (carrying a hemi or heterozygous L753F mutation in the GR) and CEM-R5C3 (characterized by impaired induction of GR expression upon Dex treatment), kindly provided by Prof. R. Kofler [60,61]. Cell lines were maintained in RPMI-1640 medium (Gibco) supplemented with 10% fetal calf serum (Greiner Bio-One) and 100 units/mL penicillin G, and 100 μg/mL streptomycin sulfate (Gibco). Cultures were refreshed twice weekly.

### 4.2. Exposure to Spliceosome Modulators

Meayamycin B (MAMB) was kindly provided by Prof. K. Koide (Department of Chemistry, University of Pittsburgh, USA), while pladienolide B (Plad-B) was purchased from Cayman Chemical Company (Ann Arbor, USA). Dex was purchased from Centrafarm (Etten-Leur, The Netherlands). Exponentially growing cell lines were seeded at a density of 0.1 × 10^6^ cells/mL and exposed to Plad-B for 72 h. Cryopreserved primary ALL samples were thawed and seeded at a density of 2 × 10^6^ cells/mL [62] and exposed to Plad-B, Dex or the combination of both. Flow cytometry-based apoptosis assay combined with immunophenotyping was performed after 72 h drug incubation. For cell lines, the drug treatment was followed by RNA extraction (after 6 h incubation), flow cytometry-based apoptosis assay, proliferation assay and cell cycle analysis assay. 

### 4.3. RNA Isolation and Reverse Transcription

Total RNA was extracted from primary ALL patient samples and cell lines (exposed or unexposed to Plad-B) using the RNeasy mini kit (Qiagen). For PCR analysis reverse transcription was carried out using 1 µg RNA and M-MLV reverse transcriptase (Invitrogen) in a reaction buffer containing random hexamer primers, dNTPs (Roche), and a ribonuclease inhibitor RNAsin (Promega).

### 4.4. RNA Sequencing

RNA integrity of isolated total RNA was assessed with RNA 6000 Nano Kit on a 2100 Bioanalyzer system (Agilent Technologies). Sequencing libraries were prepared with the Illumina TruSeq Stranded mRNA Library Prep LT Kit (RS-122-2201) and Agencount AMPure XP beads (Beckman Coulter). cDNA library size and concentration were measured by Bioanalyzer.

Single-end, 100 bp-reads were obtained from HT-v4-SR100 Chip (8 lanes) on Illumina HiSeq 2500 System. The sequencing reaction yielded 22.3 ± 5.1 (average ± SD) million passing-filter raw reads/sample (raw data are deposited in the GEO database with accession number: GSE133499). The bioinformatic pipeline for data analysis was described previously [37] and summarized in the Supplemental Methods. 

### 4.5. PCR Analysis of Splicing Events

Semi-quantitative PCR validation of selected candidate splicing events detected by rMATS was performed by using primers listed in Table A3. PCR was carried out with 2× ReddyMix PCR master mix (Thermo Fisher Scientific) [63] and the PCR products were resolved on 2% agarose gels with ethidium bromide. Digital images of gels were processed with ImageJ Software (U.S. National Institutes of Health) to calculate the intensity of each band. PCR ratio is calculated with the formula:WT/(WT + ES)(1)

### 4.6. Flow Cytometry-Based Analyses

For the proliferation and apoptosis assay, cells were first stained with 7-AAD (Via-Probe^TM^, BD Bioscience). Subsequently, for assessment of apoptosis induction, 7-AAD-stained cells were further labeled with FITC-conjugated Annexin-V antibody (Apoptest^TM^, VPS Diagnostic) in the Annexin binding buffer (Ref. v13246, Thermo Fisher Scientific). For proliferation assay total count of 7-AAD-negative cells was determined by using Flow-Count^TM^ Fluorospheres (ref.7547053, Beckman Coulter).

For cell cycle analysis the cells were permeabilized in 70% ethanol, followed by 30 min incubation with RNAse A (100 μg/mL, Qiagen) and subsequent staining with propidium iodide (Thermo Fisher Scientific).

In all assays, fluorescence was measured using BD FACS Canto II and Celesta flow cytometers (BD Bioscience). Analysis was performed using BD FACS Diva software version 8.0.1.1. Details of the flow cytometry analysis performed on primary samples are described in Supplemental Methods.

### 4.7. MTT Growth Inhibition and Cytotoxicity Assay

Growth inhibitory effects of MAMB/Plad-B in ALL cell lines, as well as primary ALL samples, were determined after a continuous exposure using the colorimetric MTT dye reduction assay as described previously [62,64]. ALL cell lines were incubated for 72 h, while primary ALL and non-malignant samples for 96 h. In combination experiments, a fixed concentration of Plad-B was combined with a dilution range of Dex. Fractional effect analysis was performed using the CalcuSyn software (Version 1.1.1, 1996, Biosoft), as described previously [65,66].

## 5. Conclusions

In conclusion, we demonstrated that BCP-ALL and T-ALL cells have distinct splicing landscape and GC resistance was associated with lineage-specific differential splicing events. Global splicing alterations affected GC resistance-relevant processes, including survival signaling, energy metabolism and ROS-induced stress response. Finally, we show that targeting splicing with small molecule splicing modulators constitutes an attractive treatment strategy that can potentially sensitize intrinsically more GC-resistant T-ALL to treatment regimens including GC.

## Figures and Tables

**Figure 1 cancers-12-00723-f001:**
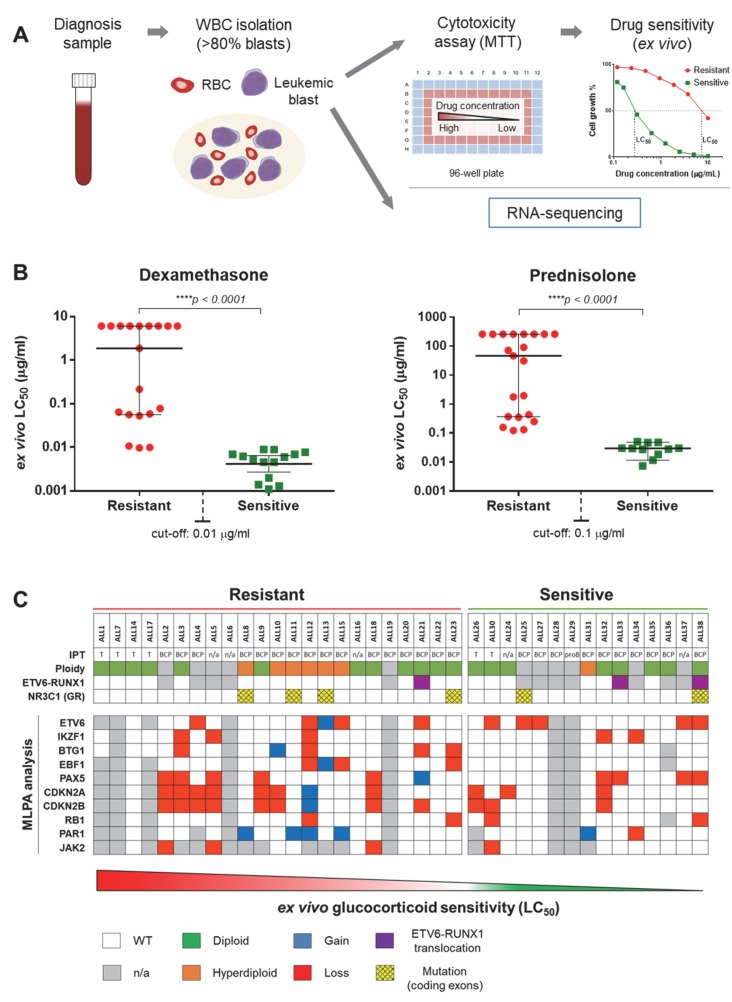
Overview of de novo pediatric acute lymphoblastic leukemia (ALL) patient cohort. (**A**) Schematic representation of study design. Primary childhood leukemia samples (from both peripheral blood and bone marrow) were collected at diagnosis and processed for white blood cell (WBC) isolation. Samples with blast populations >80% were tested for ex vivo cytotoxicity (MTT) assays and processed for RNA sequencing. (**B**) Ex vivo glucocorticoid (GC) sensitivity levels. Isolated blasts were treated with Dex (concentration range: 0.0002–6.1 μg/mL) and Pred (concentration range: 0.007–260 μg/mL). After 96 h, the samples were measured by MTT assay and the median lethal concentration (LC_50_—the concentration of the drug that kills 50% of cells as compared to the control) and 95% confidence interval (CI) was determined. GC sensitivity cut-off was set at 0.01 μg/mL for Dex and 0.1 μg/mL for Pred. (**C**) Chromosomal and genetic alterations. The table contains data concerning ploidy, immunophenotype, ETV6-RUNX1 translocations, NR3C1 (GR) mutations and detection of gains and losses of several genes relevant for ALL pathogenesis and GC resistance through MLPA analysis. MLPA: multiplex ligation-dependent probe amplification, IPT: immunophenotype, GR: glucocorticoid receptor.

**Figure 2 cancers-12-00723-f002:**
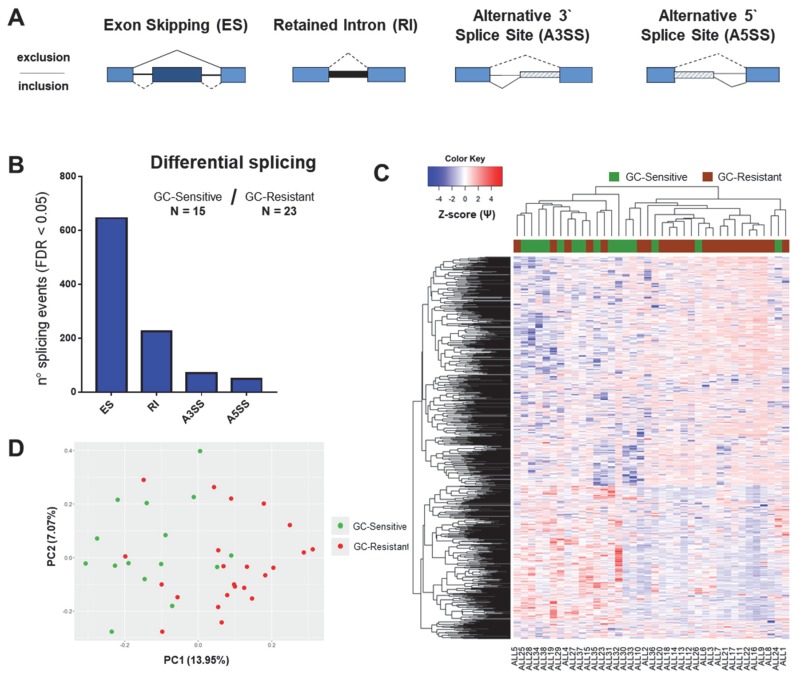
Global splicing profiles related to GC resistance in the total cohort of childhood ALL patients. The figure depicts the results of differential splicing analysis performed on 15 GC-sensitive versus 23 GC-resistant childhood ALL samples using the rMATS algorithm. (**A**) Schematic representation of alternative splicing event types as detected by rMATS. ES: exon skipping (dark blue exon can be excluded from the mRNA transcript); RI: retained intron (thick black line represents the intron that is included in the mRNA); A3SS: alternative 3’ splice site; A5SS: alternative 5’ splice site (depending on the position of the splice site, part of introns are included in the mRNA and depicted as striped boxes). (**B**) The number of significant splicing (FDR < 0.05) events per each type. (**C**) Hierarchical clustering performed using all significant differential splicing events. Inclusion levels (or Percentage Spliced-In, Ψ) per each event were generated by rMATS, Z-score-normalized and plotted as a heatmap. The colored bar over the heatmap represents the sensitivity to GCs (green—GC-sensitive samples, brown—GC-resistant). (**D**) Principal component analysis (PCA) plot created by using Ψ values for all significant differential splicing events.

**Figure 3 cancers-12-00723-f003:**
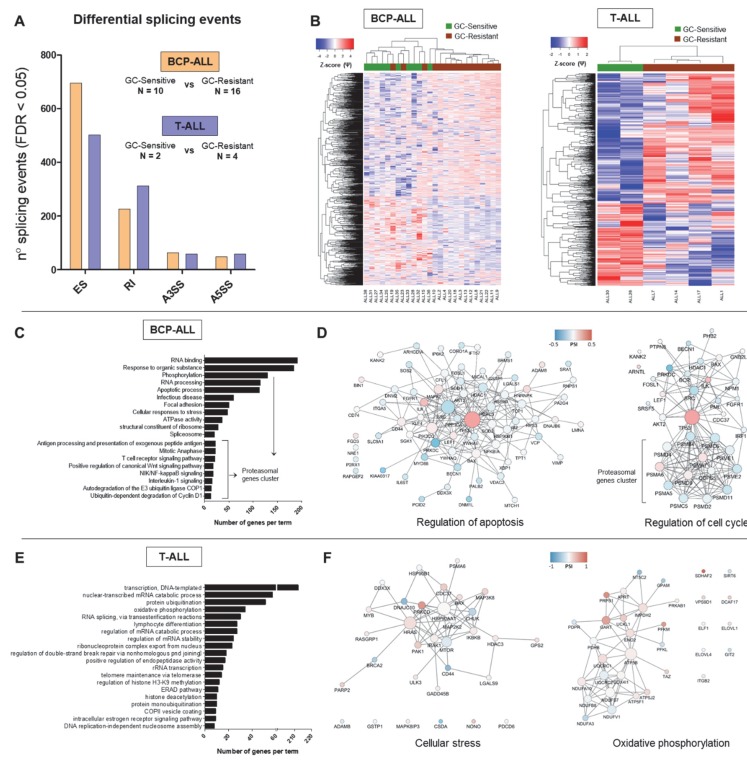
Differential splicing profiles associated with GC resistance in childhood B-cell-precursor (BCP)-ALL and T-ALL. The figure depicts the results of differential splicing analysis performed on 10 GC-sensitive versus 16 GC-resistant BCP-ALL and 2 GC-sensitive versus 4 GC-resistant T-cell ALL samples using the rMATS algorithm. (**A**) The number of significant (FDR < 0.05) events detected by comparing GC-sensitive vs. GC-resistant BCP-ALL (in orange) and T-ALL (in blue) per each type (ES—exon skipping, RI—intron retention, A3SS—alternative 3’ splice site, A5SS—alternative 5’ splice site). (**B**) Hierarchical clustering performed using all significant differential splicing events. Inclusion levels (or Percentage Spliced-In, Ψ) per each event were generated by rMATS, Z-score-normalized and plotted as a heatmap. The colored bar over the heatmap represents the sensitivity to GCs (green—GC-sensitive samples, brown—GC-resistant). (**C**,**E**) Top 20 major gene ontology (GO) terms with the largest numbers of genes in the BCP-ALL and T-ALL datasets respectively. GO search was performed in gProfiler by selecting GO, KEGG and REACTOME databases and ClueGO (Cytoscape plugin) using all genes affected by significant differential splicing events. (**D**,**F**) Gene network analysis of differentially spliced genes in relation to GC resistance in BCP-ALL and T-ALL respectively. Two representative gene networks per each dataset are shown (“regulation of apoptosis” and “regulation of cell cycle” for the BCP-ALL, “cellular stress” and “oxidative phosphorylation” for the T-ALL). Gene networks were obtained using the STRING tool and visualized in Cytoscape. The color represents the Ψ and the size of the nodes represents degree (number of connection to neighboring nodes).

**Figure 4 cancers-12-00723-f004:**
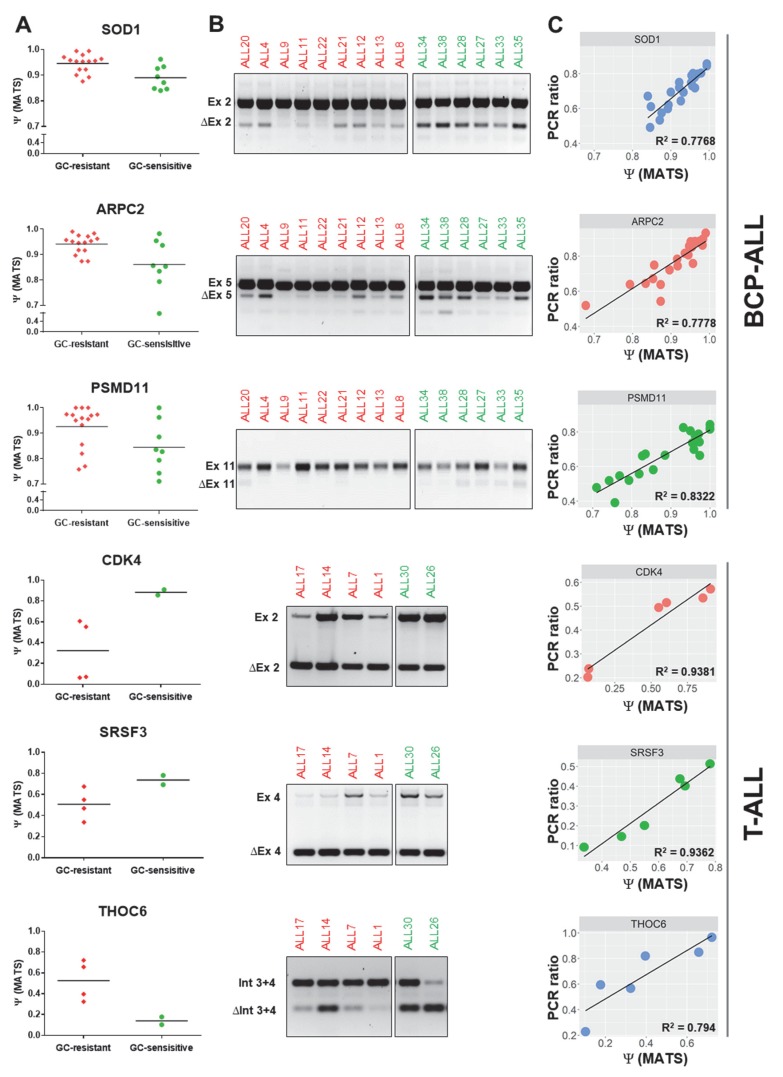
PCR validations of differential splicing events predicted by the rMATS algorithm. The PCR validations were performed for 13 differential splicing events identified in BCP-ALL and 19 differential splicing events identified in T-ALL (Figure A1). The figure depicts 3 selected genes per each subtype, including splicing regulators and GC resistance-related genes. (**A**) Ψ values generated by rMATS in GC-resistant and GC-sensitive samples (FDR < 0.05). (**B**) Electrophoresis gels illustrating the results of PCR validations for selected differential splicing events identified by rMATS. (**C**) Association between rMATS-calculated and PCR-derived Ψ values (PCR ratio) was evaluated per each validated event using the linear regression model. *Ex:* exon inclusion; *∆Ex:* exon skipping; *Int:* retained intron; *∆Int*: intron exclusion.

**Figure 5 cancers-12-00723-f005:**
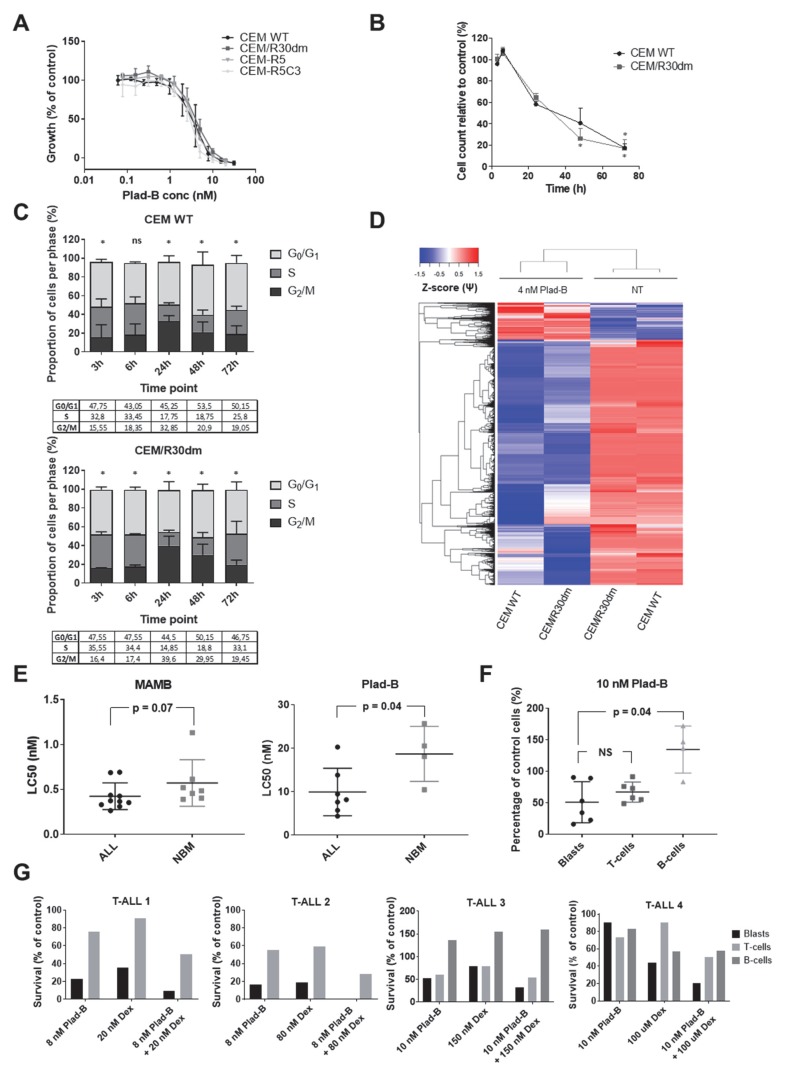
Response of (GC-resistant) T-cell ALL cells to splicing modulation. The figure illustrates the effects of splicing modulation on T-ALL cell lines and primary childhood samples as well as non-malignant specimens. (**A**) Response to Plad-B of GC-sensitive CCRF-CEM-WT cells as well as its GC-resistant sublines: CEM-R30dm, CEM-R5 and CEM-R5C3 in a 72 h MTT assay. The plot depicts the mean ± SD of at least 3 independent experiments. (**B**,**C**) Time-dependent inhibition of proliferation (**B**) and cell cycle arrest (**C**) induced by treatment with 4 nM Plad-B in CEM-WT and CEM-R30dm. The panel depicts the mean ± SD of 2 independent experiments. T-test was used for (**B**); asterisks in (**B**,**C**) indicate statistical significance (*p* < 0.05); in (**B**), statistical significance for CEM-WT is indicated above the plotted line and for CEM-R30dm below the plotted line. Chi-square test was used for (**C**). (**D**) Hierarchical clustering of significant differential splicing events between untreated (NT) and 4 nM Plad-B-treated CEM-WT and CEM-R30dm cells. Inclusion levels (Ψ) were generated by rMATS Z-score-normalized and plotted as a heatmap. (**E**) The effect of MAMB and Plad-B treatment on primary samples of childhood ALL patients and non-malignant cells. The first two graphs depict LC_50_ values (the concentration of the drug that kills 50% of cells as compared to the control) obtained for primary ALL samples and non-malignant bone marrow specimens in the 96 h MTT assay. (**F**) Percentage of viable cells in 3 cell subpopulations of primary childhood T-ALL samples (blast cells, mature T-cells and B-cells) upon a 72 h incubation with 10 nM Plad-B. The *p*-value was calculated using the Mann–Whitey U test. (**G**) Percentage of viable cells in 3 cell subpopulations of primary childhood T-ALL samples (blast cells, mature T-cells and B-cells) upon a 72 h incubation with Plad-B alone, Dex alone or the combination of the two. Each graph represents a single sample. The concentrations of Plad-B and Dex that were equivalent to the single drug LC_50_ values as determined in the 96 h MTT assay were used (with the exception of sample T-ALL4 for which not enough material was available to perform the MTT assay prior to flow cytometry. For this sample, the drug concentrations were selected based on the mean LC_50_ values in the total number of T-ALL samples assessed).

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
