# Peer review of "Glucocorticoid Resistant Pediatric Acute Lymphoblastic Leukemia Samples Display Altered Splicing Profile and Vulnerability to Spliceosome Modulation"

_cancers, 2020, doi:10.3390/cancers12030723_

Round 1

Reviewer 1 Report

Glucocorticoid (GC) resistance is a big cause of reduced response to chemotherapy in pediatric acute lymphoblastic leukemia (ALL). This resistance is developed through deregulated splicing of the leukemia cells. Using several RNA-seq-based differential splicing analysis and real time qPCR, distinct splicing events were pinpointed to BCP-ALL and T-ALL lines. These events associated with GC-resistance also showed involvement in RNA processing, apoptosis, cell cycle regulation, and energy metabolism. Tests on GC resistant ALL cell lines showed a sensitivity to splicing modulations, creating a new target option for GC-resistant patients. The nature of this study required the use of many RNA sequencing, qPCR, and western blot assays. These were all presented clearly. My comments are below.

Major Points:

  1. What are the connections from the splicing events to specify the energy metabolism and cell cycle regulation?
  2. In Figure 4A, there is no difference regarding to expressions of SOD1 in both GC-sensitive and GC-resistant cells. The data is different from the descriptions in Lines 192-194. It said "the expression of SOD1 was higher in sensitive cells.
  3. When the researchers went to confirm the splicing analysis from rMATS using semi-quantitative RT-PCR, the R2 values for half of the values were below .9. For example, in Line 243, the association index (R2=0.4717) is low.
  4. In Lines 266-268, the LC50 in ALL cells tended to be lower compared to non-malignant samples (P=0.07, median LC50 values 0.42 ± 0.05nM and 0.57 ± 0.1nM……). However, the P value indicates that the difference of a comparison is not significant. And the fold change (less than twofold change) is minimal.
  5. In Fig. 5A, there is no differences for Plad-B response between sensitive and resistant cells.
  6. In Lines 371-372 of Discussion Section, GC-resistant BCP-ALL cells are less prone to GC-induced ROS-mediated apoptosis due to increased level of SOD1. However, in lines 193-194, it was mentioned that SOD1 and SOD2 increased in GC-sensitive cells. How do you explain the confusion?

Minor Points:

  1. In Abstract, list the full name of BCP?
  2. Tests were conducted in a small pilot sample. The test should be expanded to make comparisons between individual and further look at the results of the modulators.
  3. In Figure 5G, no error bar in all the figures.

Author Response

Response to Reviewer 1 Comments

We thank the Reviewer for the constructive and positive remarks and we present the point-by-point replies below in red.

Major Points:

  1. What are the connections from the splicing events to specify the energy metabolism and cell cycle regulation?

Response 1:

In our study, we found splicing alterations in multiple regulators of energy metabolism and cell cycle (in BCP-ALL) in GC-resistant cell as compared to GC-sensitive cells. The observed splicing deregulation in SOD1 and SOD2 genes in BCP-ALL as well as in genes involved in oxidative phosphorylation (in T-ALL) could potentially mitigate GC-induced generation of ROS and allow GC-resistant cells to evade apoptosis. Increased inclusion of the specific exons in SOD1/2 could result in their increased metabolic activity in GC-resistant cells and protect them from ROS-induced apoptosis. Similarly, in T-ALL we observed predominantly retained introns in genes involved in oxidative phosphorylation. As intron retentions most commonly introduce premature stop codons, these events are likely to lead to decreased expression (and consequently function) of these enzymes abrogating GC-induced switch to oxidative phosphorylation and the resultant inhibition of ROS-induced apoptosis. Similarly, TP53 splice variant β (predominantly expressed in GC-resistant cells) is likely to affect TP53α-mediated cell cycle and apoptosis regulation differently as compared to the TP53γ (associated with GC-sensitivity).

We have now emphasized this better in the text on line 381 – 387: “We found that GC-resistant T-ALL cells display altered splicing (in particular intron retention) of multiple genes involved in oxidative phosphorylation, in particular components of ubiquitinol-cytochrome c reductase and NADH dehydrogenase complex (which promotes DNA damage-induced apoptosis through production of reactive oxygen species - ROS) [60]. As intron retention often generates premature stop codons and transcript degradation, these changes are likely to result in diminished GC-induced oxidative phosphorylation and therefore suppress production of ROS and consequently apoptosis induction.” and line 365-368: “For instance, TP53 spliced variants β and γ have previously been reported to alter the transcriptional activity of TP53α consequently affecting cell cycle and apoptosis regulation [76]. Although functional differences between TP53β and TP53γ remain unclear, they could presumably interfere with GC-related anti-proliferative and pro-apoptotic effects.”.

  1. In Figure 4A, there is no difference regarding to expressions of SOD1 in both GC-sensitive and GC-resistant cells. The data is different from the descriptions in Lines 192-194. It said "the expression of SOD1 was higher in sensitive cells.

Response 2:

We agree with the reviewer that the difference in inclusion of exon 2 between GC-resistant and GC-sensitive samples is not large, but statistically significant according to the rMATS algorithm. Notably, several previous studies, including Yoshimi et al. (Coordinated alterations in RNA splicing and epigenetic regulation drive leukaemogenesis. Nature 574, 273–277 (2019); https://doi.org/10.1038/s41586-019-1618-0) and Zhang et al. (Pan-cancer analysis of clinical relevance of alternative splicing events in 31 human cancers. Oncogene 38, 6678–6695 (2019); https://doi.org/10.1038/s41388-019-0910-7), have shown that relatively small differences in splicing can be meaningful. Furthermore, skipping of exon 2 in SOD1 could possibly lead to rapid transcript degradation and therefore the actual proportion of transcripts that exclude exon 2 could be underestimated.

We would also like to thank the reviewer for pointing out that our description in the text is confusing. We have now rephrased the sentence in Line 194 to: “Both SOD1 and SOD2 showed increase in exon inclusion in GC-resistant samples.”.

  1. When the researchers went to confirm the splicing analysis from rMATS using semi-quantitative RT-PCR, the Rvalues for half of the values were below .9. For example, in Line 243, the association index (R2=0.4717) is low.

Response 3:

We concur with the reviewer that the R2 values for linear correlation between PSI and RT-PCR data may seem low. These results were obtained using a semi-quantitative RT-PCR which does not quantify transcripts with high accuracy. We reckon that these findings should be confirmed in follow-up experiments with meticulously designed, fully quantitative splice variant-specific PCRs. We have now added this remark in the discussion, Lines 336-339: “Selected splicing events were validated by RT-PCR and showed moderate to strong correlation to rMATS-derived PSI values in the majority of cases. However, due to the semi-quantitative nature of the RT-PCR method, our results should be additionally confirmed by qRT-PCR in future studies”.

  1. In Lines 266-268, the LC50 in ALL cells tended to be lower compared to non-malignant samples (P=0.07, median LC50 values 0.42 ± 0.05nM and 0.57 ± 0.1nM……). However, the P value indicates that the difference of a comparison is not significant. And the fold change (less than twofold change) is minimal.

Response 4:

We agree with the reviewer that the LC50 values for MAMB only showed borderline significance towards lower values in leukemic cells as compared to normal. However, the difference for the other splicing modulator Plad-B was significant and close to two-fold. Our statement refers to both drugs, which do show the same tendency.

These data indicate that spliceosome modulators show variable efficacy. Therefore, we suggest to also investigate more extensively their use in combination with GCs to reach an optimal therapeutic window. We now added this remark in the Discussion section, lines 421-423: “The notion that the combination of Plad-B with Dex had a more selective effects on blast cells as compared to mature lymphocytes further encourages confirmatory studies in larger samples sets and with different classes of spliceosome modulators. Given the variable potency of such compounds, future tests in combination with GCs should be carried out to determine an optimal therapeutic window.”.

  1. In Fig. 5A, there is no differences for Plad-B response between sensitive and resistant cells.

Response 5:

This is indeed true, the set of GC-resistant cell lines did not show increased resistance to splicing modulation but responded to the same degree as GC-sensitive cells. This implicates that spliceosome modulation is equally effective in both cell types, providing an attractive pharmacological opportunity for chemo-refractory patients.

We were also not expecting the GC-resistant cells to respond better to splicing modulation but rather that the combination of splicing modulation with GCs as commonly used ALL drug will cooperate in these cells, as shown in Supplementary Figure 4.

We have mentioned this more clearly in the Discussion section, lines 403-405: “Our data extend previous report showing that MAMB was able to eradicate multidrug-resistant breast cancer cells [61] and provide an attractive pharmacological opportunity for chemo-refractory patients.”  and  lines 421-423 (as reported in Response 4).

  1. In Lines 371-372 of Discussion Section, GC-resistant BCP-ALL cells are less prone to GC-induced ROS-mediated apoptosis due to increased level of SOD1. However, in lines 193-194, it was mentioned that SOD1 and SOD2 increased in GC-sensitive cells. How do you explain the confusion?

Response 6:

We thank the reviewer for pointing out this detail. Indeed, we have made an incorrect statement here and apologize for that. As can be seen in Figure 4, SOD1 in GC-resistant samples show increased levels of exon 2 inclusion. Conversely, increased exon skipping in GC-sensitive samples (represented by lower Ψ values) potentially lead to transcript degradation and decreased total gene expression. These data are supported by MASER analysis in Supplemental Figure A2 (Panel A), showing that both SOD1 and SOD2 exon skipping events could potentially affect a functional protein domain. Therefore, we have rephrased the sentence in the discussion Line 387 to: “Similarly, we found increased inclusion of specific exons in SOD1 and SOD2 genes in resistant BCP-ALL samples.” as well as the sentence in results Line 194 as indicated in Response 2 above.

Minor Points:

  1. In Abstract, list the full name of BCP?

Response 7: We have now added a full description of this abbreviation in the abstract (Line 45): “Our findings revealed for the first time markedly distinct splicing landscapes in ALL samples of (B-cell precursor) BCP-ALL and T-ALL lineages”.

  1. Tests were conducted in a small pilot sample. The test should be expanded to make comparisons between individual and further look at the results of the modulators.

Response 8: We completely agree with the reviewer and we regret we did not have access to a larger sample set at the time. We concur that future studies should validate our findings in larger sample sets and could help to predict which individuals are the most likely to respond to splicing modulation in combination with GC therapy. We have now added an additional sentence in the Discussion section, Lines 398-399, to make this limitation more clear: “Low sample numbers (in particular for T-ALL) is a limitation of the current study and therefore our findings should be further confirmed in larger datasets.”.

  1. In Figure 5G, no error bar in all the figures.

Response 9: As correctly pointed out by the reviewer, there are no error bars in this panel because each graph represents a single sample of our pilot data set. Therefore, it is very important to confirm these findings in a larger sample sets and find determinants of response of particular GC-resistant samples to the combination of splicing modulators with GCs.

Reviewer 2 Report

In this manuscript title as “Glucocorticoid resistant pediatric acute lymphoblastic leukemia samples display altered splicing profile and vulnerability to spliceosome modulation”; Sciarrillo et al exploited the primary ALL samples and cell lines to investigate the correlation between resistances to Glucocorticoid and altered splicing profile. Considering the Glucocorticoid resistance as a pertinent factor in devising the treatment strategy against ALL authors has certainly picked up a highly interesting topic with potential to directly affect the clinical management of ALL. Authors have meticulously utilized the RNA sequencing approach to understand the differential splicing events ALL clinical samples in terms of GC sensitivity.   Most of the experiments in this manuscript are well designed however; still, I expect more clarification from the authors before finally recommending acceptance in Cancers.

Specific comments are as follows:

  1. In the figure 1C there is no signatory difference in chromosomal aberrations or in gene list in the MLPA analysis. What is the conclusion of their MLPA analysis or from figure 1C. Were they able to characterize and differentiate between GC resistant and sensitive cells in terms of some known markers?
  2. In their evaluation of the therapeutic potential of splicing modulation in drug-resistant ALL using pladienolide B (Figure 5), the difference in cell proliferation inhibition and cell cycle arrest is very nominal. In such cases, authors should provide statistical analysis to assess if these changes are significant or not.

Author Response

We thank the Reviewer for the constructive and positive remarks and we present the point-by-point replies below in red.

  1. In the figure 1C there is no signatory difference in chromosomal aberrations or in gene list in the MLPA analysis. What is the conclusion of their MLPA analysis or from figure 1C. Were they able to characterize and differentiate between GC resistant and sensitive cells in terms of some known markers?

Response 1:

The reviewer is right that there is no difference in chromosomal and structural aberrations between GC-sensitive and GC-resistant. With this analysis we were aiming to examine whether our GC-sensitive and GC-resistant patient groups differ significantly with respect to genetic alterations which could confound our analysis. We found no differences between our groups.

  1. In their evaluation of the therapeutic potential of splicing modulation in drug-resistant ALL using pladienolide B (Figure 5), the difference in cell proliferation inhibition and cell cycle arrest is very nominal. In such cases, authors should provide statistical analysis to assess if these changes are significant or not.

Response 2:

We thank the Reviewer for pointing this out. We now performed statistical testing and indicated the significant datapoints in Figure 5B and 5C. We also amended the figure legend in Lines 300-303 as follows: “T-test was used for panel B; asterisks in panels B and C indicate statistical significance (p-value <0.05); in panel B statistical significance for CEM-WT is indicated above the plotted line and for CEM-R30dm below the plotted line. Chi-square test was used for Panel C.”

Reviewer 3 Report

    This manuscript reports  the global splicing alteration profiles in BCP-ALL and T-ALL and effect of splicing modulation   on Glucocorticoid (GC) resistance. There are several questions to be addressed.

  1. SRSF3 and SRPK2 were found alternatively spliced in the T-ALL dataset and SRSF5 and SRSF7 were specific for the BCP-ALL group. Alteration of splicing or expression of these splicing factors can cause further alteration of splicing unrelated to GC sensitivity.
  2. In the current study, a very small pilot sample set for T-ALL is analyzed, which limits the conclusions.
  3. Figure 3.(A) The numbers of significant (FDR < 0.05) events per each type (ES – exon skipping, RI – intron retention, A3SS – alternative 3’ splice site, A5SS – alternative 5’ splice site) at T-ALL and BCP-ALL are shown. The comparison of GC-sensitive vs. GC-resistant at T-ALL and BCP-ALL are not shown.
  4. Why are three GC-resistant BCP-ALL patients shown similar pattern of splicing alteration to GC-sensitive BCP-ALL group in figure 3B?
  5. In Supplemental Figure A1. (B) It is not clearly shown the comparison between T-All to BCP-ALL. Only one group data of either T-ALL or BCP-ALL are shown.

Author Response

We thank the Reviewer for the constructive and positive remarks and we present the point-by-point replies below in red.

  1. SRSF3 and SRPK2 were found alternatively spliced in the T-ALL dataset and SRSF5 and SRSF7 were specific for the BCP-ALL group. Alteration of splicing or expression of these splicing factors can cause further alteration of splicing unrelated to GC sensitivity.

Response 1:

This comment is well taken. A recent study by Black et al. (Aberrant splicing in B-cell acute lymphoblastic leukemia, Nucleic Acids Research, 46 (21), 11357–11369 (2018); https://doi.org/10.1093/nar/gky946) showed that splicing deregulation of the factor HNRNPA1 can drive widespread changes in B-ALL splicing and contribute to disease pathogenesis.

In our study, SRSF3 and SRPK2 were alternatively spliced in GC-resistant as compared to GC-sensitive T-ALL while SRSF5 and SRSF7 were alternatively spliced in GC-resistant as compared to GC-sensitive BCP-ALL. We hypothesize  that differential splicing of these splice factors in the context of GC-resistance might also contribute to the global changes in splicing we have demonstrated.

To better elaborate on this, we added few sentences in the Discussion section, Lines 349-354:

“In particular, we want to emphasize that specific classes of splicing regulators are associated with specific immunophenotypes. Hence, distinct regulatory splicing pathways/networks related to GC-resistance should be investigated within each subtype, to avoid interference of different splicing related to processes such as cell differentiation. Future studies should evaluate to which degree differential splicing of these particular splicing factors contribute to the globally altered splicing profiles in GC-resistant cells.”.

  1. In the current study, a very small pilot sample set for T-ALL is analyzed, which limits the conclusions.

Response 2:

We agree with the reviewer that our samples set (in particular for T-ALL) is relatively small due to limited numbers of fully annotated patient samples. We have now added an additional sentence in the Discussion section, Lines 398-399, to refer to this limitation: “Low sample numbers (in particular for T-ALL) is a limitation of the current study and therefore our findings should be further confirmed in larger datasets.”.

  1. Figure 3.(A) The numbers of significant (FDR < 0.05) events per each type (ES – exon skipping, RI – intron retention, A3SS – alternative 3’ splice site, A5SS – alternative 5’ splice site) at T-ALL and BCP-ALL are shown. The comparison of GC-sensitive vs. GC-resistant at T-ALL and BCP-ALL are not shown.

Response 3:

The number of differential splicing events found in the comparison of BCP-ALL vs T-ALL is depicted in Supplemental Figure 1B while Figure 3A illustrates the number of significant differential splicing events found in the comparison between GC-sensitive and resistant samples (as indicated in the top-right corner of Figure 3, panel A) in each subtype (BCP-ALL in orange and T-ALL in blue) separately. In order to avoid confusion, we extended the Figure 3A text legend in Lines 223-224: “The number of significant (FDR < 0.05) events detected by comparing GC-sensitive vs GC-resistant BCP (in orange) and T-ALL (in blue) per each type (ES – exon skipping, RI – intron retention, A3SS – alternative 3’ splice site, A5SS – alternative 5’ splice site).

  1. Why are three GC-resistant BCP-ALL patients shown similar pattern of splicing alteration to GC-sensitive BCP-ALL group in figure 3B?

Response 4:

As reported in Supplemental Data S1, the three GC-resistant samples the Reviewer refers to are ALL23 (LC50 Pred: 0.12 µg/mL; LC50 Dex 0.01 µg/mL), ALL19 (LC50 Pred: 0.16 µg/mL; LC50 Dex 0.06 µg/mL) and ALL15 (LC50 Pred: 1.75 µg/mL; LC50 Dex: missing data). These LC50 values indicate that the samples in question have moderate levels of ex-vivo GC resistance, close to the cut-off values indicated in Figure 1B (LC50 Pred: 0.1 μg/mL; LC50 Dex: 0.01 μg/mL). Therefore, their splicing profiles can be similar to GC-sensitive samples and cluster accordingly.

  1. In Supplemental Figure A1. (B) It is not clearly shown the comparison between T-All to BCP-ALL. Only one group data of either T-ALL or BCP-ALL are shown.

Response 5:

We thank the reviewer for pointing out this apparent confusion. In this Supplemental figure 1B-E we show the results of differential splicing analysis comparing all the BCP-ALL samples to all the T-ALL samples regardless of their GC-sensitivity status. We have now rephrased our legend to explain more clearly how we performed the analysis (Lines 556-559): “The figure depicts gene expression profiles and differential splicing analysis comparing 6 T-ALL to 26 BCP-ALL samples. This analysis was performed to evaluate differences in splicing due to the lineage of cells (all T-ALL samples were compared to all BCP-ALL samples).”.

Round 2

Reviewer 1 Report

The writers addressed all of my questions stated in the first review. The corrections made cleared up much confusion previously seen in the first draft. While the writers stated in the response comments that the differences in expression of SOD1 is significant based on the rMATS algorithm, the visual aspect of the graph showing similar levels weakens the graph. I think a note in the discussion or a small introduction of the rMATS algorithm in section "Results 2.1" would eliminate any confusion.
